# Homocouplings of Sodium Arenesulfinates: Selective Access to Symmetric Diaryl Sulfides and Diaryl Disulfides

**DOI:** 10.3390/molecules27196232

**Published:** 2022-09-22

**Authors:** Xin-Zhang Yu, Wen-Long Wei, Yu-Lan Niu, Xing Li, Ming Wang, Wen-Chao Gao

**Affiliations:** 1Department of Chemistry and Chemical Engineering, Taiyuan Institute of Technology, Taiyuan 030008, China; 2Department of Biomedical Engineering, Taiyuan University of Technology, Taiyuan 030024, China; 3School of Chemistry and Molecular Engineering, East China Normal University, 3663, Shanghai 200062, China

**Keywords:** sodium arenesulfinates, homocoupling, symmetric, diaryl sulfides, diaryl disulfides

## Abstract

Symmetrical diaryl sulfides and diaryl disulfides have been efficiently and selectively constructed via the homocoupling of sodium arenesulfinates. The selectivity of products relied on the different reaction systems: symmetrical diaryl sulfides were predominately obtained under the Pd(OAc)_2_ catalysis, whereas symmetrical diaryl sulfides were exclusively yielded in the presence of the reductive Fe/HCl system.

## 1. Introduction

Symmetrical sulfides [1,2,3] and disulfides [4,5,6,7] are ubiquitous structural motifs, and their corresponding derivatives have found prevalent existence in many biologically active molecules, pharmaceuticals, ligands, functional materials, and natural products. Owing to their importance, various synthetic methodologies have been developed for the preparation of these two classes of sulfur compounds [8,9,10,11,12,13,14,15,16,17,18,19,20,21,22,23]. Although the cross-coupling reactions of various sulfur surrogates with other aromatic reagents, such as aromatic halides, arenediazonium salts, and others, are generally efficient for the construction of these two symmetric structures [24,25,26,27,28,29,30,31,32,33,34,35,36], the homocoupling of arylsulfonyl derivatives [37,38] and thiols [39,40,41,42,43,44,45,46,47,48] has proven to be the most straightforward and convenient strategy in terms of simplicity. Despite these major advances, the utility of environmentally friendly sulfur sources for symmetric sulfides and disulfides is still highly desirable.

Due to their stable, greener, and inexpensive features, sodium sulfinates have been utilized as ideal sulfur donors and widely applied as the coupling partners in C-S cross-coupling reactions, such as sulfonylation [49,50], thiosulfonylation [51], sulfinylation [52,53], and sulfenylation [54,55,56,57,58,59]. Especially by reductive coupling, the sodium sulfinates could serve as sulfenylation reagents for the synthesis of unsymmetric sulfides under Pd, Cu, or I_2_ catalysts (Figure 1a) [54,55,56,57,58,59]. However, the application of sodium sulfinates for the construction of symmetric disulfides by homocoupling reactions was still less explored. On the other hand, although several reaction systems, such as EtOP(O)H_2_, TiCl_4_/Sm, WCl_6_/NaI, WCl_6_/Zn, MoCl_6_/Zn, Cp_2_TiCl_2_/Sm, and Silphos, have been reported to mediate the conversion of sodium arenesulfinates into symmetric diaryl disulfides by reductive coupling (Figure 1b) [60,61,62,63], they generally suffered from limited substrate scope, expensive reagents, or complicated procedures. Herein, we would like to report an efficient strategy for the selective synthesis of symmetrical diaryl sulfides and diaryl disulfides using sodium sulfinates as sulfenylation reagents through homocoupling.

## 2. Results and Discussion

Our initial study started with sodium phenylsulfinate **1a** as the model substrate to explore the formation of diphenylsulfide **2a**. First, by screening different solvents, NMP was proven to be the most effective out of the others, such as DMF and DMSO (Table 1, entry 3 vs. 1−2). The catalyst played a decisive role in this reaction: among the common metal catalysts, Pd(OAc)_2_ was the best one to afford diphenylsulfide **2a** in 47% yield (entry 3 vs. 4–6). With CuI and Ni(OAc)_2_ as catalysts, the product **2a** was not detected at all (entries 4 and 5). Only a trace amount of **2a** was observed when FeCl_3_ was employed (entry 6). The yield could be improved to 60% by increasing reaction temperature to 150 °C (entry 7 vs. 3 and 8). Most importantly, a decrease of catalyst loading to 2 mol% could further increase the yield to 89%, and no better result was observed by continuously reducing the catalyst loading (entry 10 vs. 7, 9–11). The desired **2a** was not detected when sodium benzenesulfinate was replaced with benzenesulfinic acid in the presence of NaOH under the same conditions (entry 12).

Having built the optimal conditions for the construction of diphenylsulfide **2a**, we turned our attention to explore the generality of sodium sulfinates. As shown in Table 2, a variety of substrates could undergo the homocoupling to afford symmetrical diaryl sulfides with high chemoselectivity. It was found that sodium benzenesulfinates with electron-donating groups such as 4-methyl, 3-methyl, 2-methyl, 4-methoxyl, 3-methoxyl, 2-methoxyl, 4-isopropyl, and 4-*tert*-butyl on the phenyl ring gave the corresponding products **2b**–**2i** in good yields. Electron-withdrawing groups, such as F, Cl, Br, and NO_2_, were also well-tolerated to provide the desired products **2j**–**2s** in moderate to good yields that were somewhat lower than the electron-donating groups offered. To our delight, the intramolecular formation of sulfides was also tried, and the desired dibenzothiophene **2t** was produced in a 51% yield.

During our studies on the synthesis of diphenylsulfide **2a**, 1,2-diphenyldisulfide **3a** was accidentally detected when using CuI as a reductant. This discovery encouraged us to search for optimal conditions for the reductive coupling of sodium benzenesulfinate for 1,2-diphenyldisulfide **3a**. Fortunately, using Fe/HCl as the reductive system, diphenyldisulfide **3a** was isolated as the major product. Subsequently, the investigation of the concentration of hydrochloric acid revealed that increasing the concentration led to the higher yield, and 12 mol/L of hydrochloric acid gave up to 96% yield (Table 3, entry 4 vs. 1−3). The highest yield was obtained when 4.0 equiv. of HCl was used (Table 2, entry 4). Increasing or decreasing the amount resulted in lower yields (entries 5 and 6 vs. 4). It was found that 2.0 equiv. of Fe was suitable for this transformation, and other amounts did not improve the yield further (entry 4 vs. 7 and 8). More notably, the similar high yield was provided when the time was shortened to 9 h (entry 9). However, sodium benzenesulfinate generated in situ by the reaction of NaOH and the equivalent of 4-methylbenzenesulfinic acid only afforded the target product (**3b**) in a 59% yield (entry 10).

With the optimized reaction conditions in hand, we next focused on the evaluation of the scope of the coupling partner to symmetric disulfides, and the results are summarized in Table 4. To our delight, it was found that the reaction could be compatible with a broad range of functional groups, furnishing the corresponding products in good to excellent yields. Although various functional groups, including electronically diverse (**3a**–**3u**) and sterically hindered (**3d**, **3g**, **3h**, **3i**, **3l,** and **3r**) ones are readily tolerated, some substantial influence of electronic properties and steric hindrance of the substituents was observed. The substrates possessing an electron-rich group (Me, MeO, *i*-Pr, and *t*-Bu) showed higher yields than those bearing an electron-poor group (F, Cl, Br, and CF_3_) (**3b**–**3i** vs. **3j**–**3l**, **3p**–**3s,** and **3u**). Among substrates, sodium *ortho*-substituted arylsulfinates, which are sterically hindered, gave relatively lower yields (**3d** vs. **3b**, **3c**, and **3g** vs. **3e**, **3f**, and **3l** vs. **3j**, **3k**, and **3r** vs. **3p** and **3q**). In addition, an 82% yield was obtained when sodium 2-naphthylsulfinate was employed as a substrate (**3t**). Notably, sodium 3-carboxybenzenesulfinate and sodium thiophene-2-sulfinate could be not transformed into the corresponding disulfides.

To further evaluate the utility of these two protocols, two gram-scale reactions were subsequently carried out (Figure 2). The corresponding products **2a** and **3a** could be afforded in 88% and 94% yields in a 10 mmol scale, respectively, demonstrating the practicability of the present methodology.

To elucidate the reaction mechanism for the homocoupling of sodium arylsulfinates, several control experiments were conducted (Figure 3). The formation of both symmetric diphenyl disulfide **3a** and diphenyl sulfide **2a** was not detected after the addition of the radical scavenger 2,2,6,6-tetramethylpiperidine-1-oxyl (TEMPO, 2 equiv.) to the standard reaction systems (Figure 3a,b), indicating that both of these two transformations underwent a free-radical process. For the synthesis of **2a**, the disulfide **3a** was detected by mass spectrometry. The preparation of disulfides from sodium arylsulfinates under Pd(OAc)_2_ catalysis was also demonstrated by Xiang and co-workers [55]. In addition, the transformation from **3a** to **2a** could be successfully realized in the presence of a catalytic amount of Pd(OAc)_2_ and sodium sulfinate **1a** (Figure 3c,d).

Based on the results of the control experiments and literature reports [64,65], a plausible mechanism for the homocoupling of sodium arylsulfinate **1** to the selective access to symmetric sulfide **2** and disulfide **3** is shown in Figure 4. First, in the reductive Fe/HCl system, disulfide **3a** could be generated via the homocoupling of the thiyl radical **A**, which comes from the radical reduction of sodium phenylsulfinate **2a** (Figure 4a). Alternatively, disulfide **3a** could be also formed in the presence of catalytic Pd(OAc)_2_. After disulfide **3a** was formed, the Pd(II)-insertion to the S-S bond produced the metal-intermediate **B**, which underwent ligand exchange to form intermediate **C**. The thermal extrusion of SO_2_ of intermediate C resulted in the formation of intermediate **D** [66], which underwent the reductive elimination to give the target sulfide **2a** and regenerate Pd(0) into the next catalytic cycle (Figure 4b).

## 3. Materials and Methods

Unless otherwise indicated, all reagents and solvents were purchased from commercial sources and used without further purification. Deuterated solvents were purchased from Sigma–Aldrich(Shanghai, China.). Refinement of the mixed system was achieved through column chromatography, which was performed on silica gel (200–300 mesh) with petroleum ether (solvent A)/ethyl acetate (solvent B) gradients as elution. In addition, all yields were referred to the isolated yields (average of two runs) of the compounds, unless otherwise specified. The known compounds were partly characterized by melting points (for solid samples), 1H NMR, and compared to authentic samples or the literature data. Melting points were measured with an RD-II digital melting point apparatus (Henan, China) and were uncorrected. ^1^H NMR data were acquired on a Bruker Advance 600 MHz spectrometer (Bruker, Germany). using CDCl_3_ as solvent. Chemical shifts are reported in ppm from tetramethylsilane, with the solvent CDCl_3_ resonance as the internal standard (CDCl_3_ = 7.26). Spectra are reported as follows: chemical shift (*δ* = ppm), multiplicity (s = singlet, d = doublet, t = triplet, q = quartet, m = multiplet), coupling constants (Hz), integration, and assignment. ^13^C NMR data were collected at 100 MHz, with complete proton decoupling. The chemical shifts are reported in ppm downfield to the central CDCl_3_ resonance (δ = 77.0). High-resolution mass spectra were performed on a micrOTOF-Q II instrument (Bruker, Germany), with an ESI source.

### 3.1. Typical Procedure for Symmetric Diaryl Sulfides 2

The mixture of sodium arylsulfinate **1** (0.4 mmol) and Pd(OAc)_2_ (2 mg, 2 mol%) in NMP (1.0 mL) was stirred at 150 °C (oil bath) until the substrate was completely consumed, which was determined by TLC. Finally, the reaction mixture was purified by silica gel column chromatography (PE: EA = 40: 1) to afford the desired coupling product diarylsulfides **2**.

### 3.2. Typical Procedure for Symmetric Diaryl Sisulfides 3

The mixture of sodium arylsulfinate **1** (0.2 mmol), Fe powder (23 mg, 2.0 equiv), and HCl (12 M, 67.0 μL) in DMF (1.0 mL) was stirred at 130 °C (oil bath), until the substrate was completely consumed, which was determined by TLC. Finally, the reaction mixture was purified by silica gel column chromatography (PE: EA = 40: 1) to afford the desired coupling product diaryldisulfides **3**.

### 3.3. Gram-Scale Reaction of Sodium Benzenesulfinate to Diphenylsulfide

The mixture of sodium benzenesulfinate **1a** (1.64 g, 10 mmol) and the catalyst, Pd(OAc)_2_ (45 mg, 2 mol%) in NMP (10 mL), was stirred at 150 °C (oil bath) until the substrate was completely consumed, which was determined by TLC. Finally, the reaction mixture was purified by silica gel column chromatography to afford the coupling product diphenylsulfide **2a** (1.640 g, 88% yield).

### 3.4. Gram-Scale Reaction of Sodium Benzenesulfinate to 1,2-Diphenyldisulfane

The mixture of sodium benzenesulfinate **1a** (1.640 g, 10 mmol), Fe powder (1.150 g, 2.0 equiv.), and 12 mol/L HCl (3.35 mL) in DMF (15 mL) was stirred at 130 °C (oil bath) until the substrate was completely consumed, which was determined by TLC. Finally, the reaction mixture was purified by silica gel column chromatography to afford the coupling product 1,2-diphenyldisulfane **3a** (2.050 g, 94% yield).

### 3.5. Characterization Data for Homo-Coupling Products of Sodium Arylsulfinates

#### 3.5.1. Characterization Data for the Products of Diaryl sulfides

**Diphenyl sulfide (2a) [8]**. Colorless liquid (33.1 mg, 89% yield); *R_f_* = 0.6 (petroleum ether); ^1^H NMR (600 MHz, CDCl_3_) *δ* 7.35−7.33 (m, 4H), 7.32−7.27 (m, 4H), 7.26−7.22 (m, 2H) ppm; ^13^C{^1^H} NMR (100 MHz, CDCl_3_) *δ* 135.9, 131.2, 129.3, and 127.1; HRMS (ESI) m/z [M + H]^+^ calculated for C_12_H_11_S 187.0576, found 187.0579.

**4,4’-Dimethyldiphenyl sulfide (2b) [67]**. White solid (36.8 mg, 86% yield); mp 57−58 °C; *R_f_* = 0.6 (petroleum ether); ^1^H NMR (600 MHz, CDCl_3_) *δ* 7.23 (*dt*, *J* = 4.8, 2.4 Hz, 4H), 7.10 (*d*, *J* = 7.8 Hz, 4H), 2.33 (s, 6H) ppm; ^13^C{^1^H} NMR (100 MHz, CDCl_3_) *δ* 136.9, 132.7, 131.1, 129.9, and 21.0; HRMS (ESI) m/z [M + H]^+^ calculated for C_14_H_15_S 215.0889, found 215.0885.

**3,3’-Dimethyldiphenyl sulfide (2c) [68]**. Colorless liquid (35.5 mg, 83% yield); *R_f_* = 0.6 (petroleum ether); ^1^H NMR (600 MHz, CDCl_3_) *δ* 7.20−7.16 (m, 4H), 7.13 (*d*, *J* = 7.8 Hz, 2H), 7.05 (*d*, *J* = 7.8 Hz, 2H), and 2.31 (s, 6H) ppm; HRMS (ESI) m/z [M + H]^+^ calculated for C_14_H_15_S 215.0889, found 215.0885.

**Di-*o*-tolylsulfide (2d)** [8]. White solid (33.4 mg, 78% yield); mp 64−65 °C; *R_f_* = 0.6 (petroleum ether); ^1^H NMR (600 MHz, CDCl_3_) *δ* 7.35−7.22 (m, 2H), 7.16 (*td*, *J* = 6.0, 1.2 Hz, 2H), 7.11−7.08 (m, 2H), 7.05 (*dd*, *J* = 6.6, 1.2 Hz, 2H), and 2.38 (s, 6H) ppm; HRMS (ESI) m/z [M + H]^+^ calculated for C_14_H_15_S 215.0889, found 215.0885.

**Bis(4-methoxyphenyl)sulfide (2e) [8]**. White solid (41.8 mg, 85% yield); mp 44−46 °C; *R_f_* = 0.6 (petroleum ether); ^1^H NMR (600 MHz, CDCl_3_) *δ* 7.29−7.26 (m, 4H), 6.83 (*dt*, *J* = 9.0, 2.4 Hz, 4H), and 3.79 (s, 6H) ppm; HRMS (ESI) m/z [M + H]^+^ calculated for C_14_H_15_O_2_S 247.0787, found 247.0785.

**Bis(3-methoxyphenyl)sulfide (2f) [8]**. White solid (39.9 mg, 81% yield); mp 45−47 °C; *R_f_* = 0.6 (petroleum ether); ^1^H NMR (600 MHz, CDCl_3_) *δ* 7.24−7.18 (m, 2H), 6.96−6.92 (m, 2H), 6.90−6.88 (m, 2H), 6.81−6.77 (m, 2H), and 3.76 (s, 6H) ppm; HRMS (ESI) m/z [M + H]^+^ calculated for C_14_H_15_O_2_S 247.0787, found 247.0785.

**Bis(2-methoxyphenyl)sulfide (2g) [8]**. White solid (35.9 mg, 73% yield); mp 73−74 °C; *R_f_* = 0.6 (petroleum ether); ^1^H NMR (400 MHz, CDCl_3_) *δ* 7.28−7.21 (m, 2H), 7.06 (*dd*, *J* = 5.6, 2.0 Hz, 2H), 6.93−6.90 (*dd*, *J* = 7.6, 0.8 Hz, 2H), 6.87−6.85 (*dd*, *J* = 6.4, 1.2 Hz, 2H), and 3.87 (s, 6H) ppm; HRMS (ESI) m/z [M + H]^+^ calculated for C_14_H_15_O_2_S 247.0787, found 247.0785.

**4,4’-Diisopropyldiphenyl sulfide (2h).** White solid (45.4 mg, 84% yield); mp 73−75 °C; *R_f_* = 0.6 (petroleum ether); ^1^H NMR (600 MHZ, CDCl_3_) *δ* 7.28−7.25 (m, 4H), 7.17−7.14 (m, 4H), 2.92−2.82 (m, 2H), and 1.24 (*d*, *J* = 6.6 Hz, 12H) ppm; ^13^C{^1^H} NMR (100 MHz, CDCl_3_) *δ* 147.8, 132.8, 131.0, 127.3, 33.7, and 23.9; HRMS (ESI) m/z [M + K]^+^ calculated for C_18_H_22_SK 309.1074, found 309.1073.

**4,4’-Di-tert-butyldiphenyl sulfide (2i) [69]**. White solid (48.3 mg, 81% yield); mp 83−84 °C; *R_f_* = 0.6 (petroleum ether); ^1^H NMR (600 MHz, CDCl_3_) *δ* 7.34−7.30 (m, 4H), 7.28 (t, *J* = 1.8 Hz, 2H), 7.26 (t, *J* = 2.4 Hz, 2H), and 1.30 (s, 18H) ppm; HRMS (ESI) m/z [M + H]^+^ calculated for C_20_H_27_S 299.1828, found 299.1821.

**4,4’-Difluorodiphenyl sulfide (2j) [8]**. Colorless liquid (33.7 mg, 76% yield); *R_f_* = 0.6 (petroleum ether); ^1^H NMR (600 MHz, CDCl_3_) *δ* 7.33−7.28 (m, 4H) and 7.03−6.98 (m, 4H) ppm; HRMS (ESI) m/z [M + H]^+^ calculated for C_12_H_9_F_2_S 223.0388, found 223.0393; ^19^F NMR (376 MHz, CDCl_3_) *δ* -114.3 ppm.

**3,3’-Difluorodiphenyl sulfide(2k)** [68]. Colorless liquid (32.0 mg, 72% yield); *R_f_* = 0.6 (petroleum ether); ^1^H NMR (600 MHz, CDCl_3_) *δ* 7.33−7.26 (m, 2H), 7.16−7.11 (m, 2H), 7.06−7.01 (m, 2H), and 6.99−6.94 (m, 2H) ppm; HRMS (ESI) m/z [M + H]^+^ calculated for C_12_H_9_F_2_S 223.0388, found 223.0393; ^19^F NMR (376 MHz, CDCl_3_) *δ* -111.5 ppm.

**Bis(2-fluorophenyl)sulfide (2l)** [70]. Colorless liquid (29.8 mg, 67% yield); *R_f_* = 0.6 (petroleum ether); ^1^H NMR (600 MHz, CDCl_3_) *δ* 7.31−7.22 (m, 4H) and 7.14−7.06 (m, 4H) ppm; HRMS (ESI) m/z [M + H]^+^ calculated for C_12_H_9_F_2_S 223.0388, found 223.0393; ^19^F NMR (376 MHz, CDCl_3_) *δ* -108.7 ppm.

**Bis(4-nitrophenyl)sulfide (2m) [8]**. White solid (40.3 mg, 73% yield); mp 156−158 °C; *R_f_* = 0.5 (petroleum ether); ^1^H NMR (600 MHz, CDCl_3_) *δ* 8.06 (*dt*, *J* = 9.0, 2.4 Hz, 2H), 7.58−7.52 (m, 2H), 7.48−7.44 (m, 2H), and 7.17 (*dt*, *J* = 9.0, 2.4 Hz, 2H) ppm; HRMS (ESI) m/z [M + H]^+^ calculated for C_12_H_9_N_2_O_4_S 277.0278, found 277.0281.

**Bis(3-nitrophenyl)sulfide (2n) [71]**. White solid (37.5 mg, 68% yield); mp 42−44 °C; *R_f_* = 0.5 (petroleum ether); ^1^H NMR (400 MHz, CDCl_3_) *δ* 8.36 (t, *J* = 4.0 Hz, 2H), 8.15−8.06 (m, 2H), 7.86−7.77 (m, 2H), and 7.53 (t, *J* = 12.0 Hz, 2H) ppm; HRMS (ESI) m/z [M + H]^+^ calculated for C_12_H_9_N_2_O_4_S 277.0278, found 277.0282.

**Bis(2-nitrophenyl)sulfide (2o) [8]**. Yellow solid (34.8 mg, 63% yield); mp 123−124 °C; *R_f_* = 0.5 (petroleum ether); ^1^H NMR (400 MHz, CDCl_3_) *δ* 8.41 (*dd*, *J* = 8.4, 1.2 Hz, 2H), 8.07 (*dd*, *J* = 8.4, 1.2 Hz, 2H), 7.84−7.77 (m, 2H), and 7.61−7.54 (m, 2H) ppm; HRMS (ESI) m/z [M + H]^+^ calculated for C_12_H_9_N_2_O_4_S 277.0278, found 277.0281.

**4,4’-Dichlorodiphenyl sulfide (2p)** [8]. White solid (42.2 mg, 83% yield); mp 88−89 °C; *R_f_* = 0.6 (petroleum ether); ^1^H NMR (600 MHz, CDCl_3_) *δ* 7.30−7.27 (m, 4H) and 7.26−7.23 (m, 4H) ppm; HRMS (ESI) m/z [M + H]^+^ calculated for C_12_H_9_Cl_2_S 254.9797, found 254.9792.

**Bis(3-clorophenyl)sulfide (2q) [71]**. Colorless liquid (39.1 mg, 77% yield); *R_f_* = 0.6 (petroleum ether); ^1^H NMR (600 MHz, CDCl_3_) *δ* 7.32−7.31 (m, 2H), 7.26−7.23 (m, 4H), and 7.22−7.19 (m, 2H) ppm; HRMS (ESI) m/z [M + H]^+^ calculated for C_12_H_9_Cl_2_S 254.9797, found 254.9792.

**Bis(2-clorophenyl)sulfide (2r) [68]**. White solid (37.1 mg, 73% yield); mp 68−70 °C; *R_f_* = 0.6 (petroleum ether); ^1^H NMR (600 MHz, CDCl_3_) *δ* 7.47 (*dd*, *J* = 7.8, 1.2 Hz, 2H), 7.24 (*td*, *J* = 7.8, 1.8 Hz, 2H), 7.19 (*td*, *J* = 7.8, 1.2 Hz, 2H), and 7.14 (*dd*, *J* = 7.8, 1.8 Hz, 2H) ppm; HRMS (ESI) m/z [M + H]^+^ calculated for C_12_H_9_Cl_2_S 254.9797, found 254.9792.

**Bis(4-bromophenyl)sulfide (2s)** [8]. White solid (54.7 mg, 80% yield); mp 110−111 °C; *R_f_* = 0.6 (petroleum ether); ^1^H NMR (600 MHz, CDCl_3_) *δ* 7.45−7.40 (m, 4H) and 7.21−7.16 (m, 4H) ppm; HRMS (ESI) m/z [M + H]^+^ calculated for C_12_H_9_Br_2_S 342.8786, found 342.8784.

**Dibenzo[b,d]thiophene (2t)** [72]. white solid (38 mg, 51% yield); m.p. 95–96 °C. Rf = 0.8 (PE/EA = 20:1); ^1^H NMR (400 MHz, CDCl_3_) *δ* 8.17 (m, 2H), 7.87 (*dd*, 2 H, J = 8.0, 4.0 Hz), and 7.47 (m, 4 H). The NMR data were consistent with the previous report (see spectra at Appendix A).

#### 3.5.2. Characterization Data for the Products of Diaryl Disulfides

**Diphenyl disulfide (3a)** [73]. White solid (21.0 mg, 96% yield); mp 61−62 °C; R*_f_* = 0.6 (petroleum ether); ^1^H NMR (600 MHz, CDCl_3_) *δ* 7.54−7.45 (m, 4H), 7.33−7.26 (m, 4H), and 7.25−7.19 (m, 2H) ppm; ^13^C{^1^H} NMR (100 MHz, CDCl_3_) *δ* 137.0, 129.0, 127.5, and 127.1; HRMS (ESI) m/z [M]^+^ calculated for C_12_H_10_S_2_ 218.0224, found 218.0217.

**4-Methylphenyl disulfide (3b)** [73]. White solid (22.9 mg, 93% yield); mp 47−48 °C; *R_f_* = 0.6 (petroleum ether); ^1^H NMR (600 MHz, CDCl_3_) *δ* 7.39 (*d*, *J* = 8.4 Hz, 4H), 7.11 (*d*, *J* = 7.8 Hz, 4H), and 2.32 (s, 6H) ppm; ^13^C{^1^H} NMR (100 MHz, CDCl_3_) *δ* 137.4, 133.9, 129.8, 128.6, and 21.0; HRMS (ESI) m/z [M]^+^ calculated for C_14_H_14_S_2_ 246.0537, found 246.0517.

**3-Methylphenyl disulfide (3c)** [73]. White solid (22.1 mg, 90% yield); mp 112−114 °C; *R_f_* = 0.6 (petroleum ether); ^1^H NMR (600 MHz, CDCl_3_) *δ* 7.31 (*d*, *J* = 7.2 Hz, 4H), 7.19 (t, *J* = 7.4 Hz, 2H), 7.04 (*d*, *J* = 7.2 Hz, 2H), and 2.32 (s, 6H) ppm; HRMS (ESI) m/z [M]^+^ calculated for C_14_H_14_S_2_ 246.0537, found 246.0519.

**Di(*o*-methylphenyl)disulfide (3d)** [73]. White solid (20.9 mg, 85% yield); mp 40−42 °C; R*_f_* = 0.6 (petroleum ether); ^1^H NMR (600 MHz, CDCl_3_) *δ* 7.24−7.21 (m, 2H), 7.19−7.15 (m, 2H), 7.11−7.07 (m, 2H), 7.06 (*dd*, *J* = 7.8, 1.8 Hz, 2H), and 2.37 (s, 6H) ppm; HRMS (ESI) m/z [M]^+^ calculated for C_14_H_14_S_2_ 246.0537, found 246.0517.

**Di(4-methoxyphenyl)disulfide (3e) [34]**. White solid (25.6 mg, 92% yield); mp 45−47 °C; *R_f_* = 0.6 (petroleum ether); ^1^H NMR (600 MHz, CDCl_3_) *δ* 7.31−7.27 (m, 4 H), 6.87−6.81 (m, 4 H), and 3.79 (s, 6H) ppm; HRMS (ESI) m/z [M]^+^ calculated for C_14_H_14_O_2_S_2_ 278.0435, found 278.0423.

**Di(3-methoxyphenyl)disulfide (3f) [34]**. White solid (24.5 mg, 88% yield); mp 106−108 °C; *R_f_* = 0.6 (petroleum ether); ^1^H NMR (400 MHz, CDCl_3_) *δ* 7.18−7.11 (m, 2H), 7.03−6.99 (m, 2H), 6.88−6.81 (m, 2H), 6.73−6.67 (m, 2H), and 3.69 (s, 6H) ppm; HRMS (ESI) m/z [M]^+^ calculated for C_14_H_14_O_2_S_2_ 278.0435, found 278.0423.

**Di(2-methoxyphenyl)disulfide (3g) [34]**. White solid (23.4 mg, 84% yield); mp 120−121 °C; *R_f_* = 0.6 (petroleum ether); ^1^H NMR (600 MHz, CDCl_3_) *δ* 7.53 (*dd*, *J* = 6.6, 1.2 Hz, 2 H), 7.21−7.16 (m, 2 H), 6.93−6.88 (m, 2 H), 6.86 (*d*, *J* = 7.8 Hz, 2 H), and 3.90 (s, 6 H) ppm; HRMS (ESI) m/z [M]^+^ calculated for C_14_H_14_O_2_S_2_ 278.0435, found 278.0427.

**1,2-bis(4-isopropylphenyl)disulfane (3h) [73]**. White solid (27.5 mg, 91% yield); mp 79−81 °C; *R_f_* = 0.6 (Petroleum ether); ^1^H NMR (600 MHz, CDCl_3_) *δ* 7.44 (*dt*, *J* = 8.4, 4.8 Hz, 4H), 7.17 (*dt*, *J* = 7.8, 4.2 Hz, 4H), 2.93−2.86 (m, 2H), and 1.24 (*d*, *J* = 7.2 Hz, 12H) ppm; HRMS (ESI) m/z [M + H]^+^ calculated for C_18_H_23_S_2_ 303.1236, found 303.1226.

**Bis(4-tert-butylphenyl) disulfide (3i) [74]**. White solid (28.7 mg, 87% yield); mp 88−89 °C; *R_f_* = 0.6 (petroleum ether); ^1^H NMR (600 MHz, CDCl_3_) *δ* 7.46 (*d*, *J* = 9.0 Hz, 4H), 7.34 (*d*, *J* = 8.4 Hz, 4H), and 1.31 (s, 18H) ppm; HRMS (ESI) m/z [M]^+^ calculated for C_20_H_26_S_2_ 330.1476, found 330.1462.

**Bis(4-fluorophenyl) disulfide (3j) [34]**. White solid (21.8 mg, 86% yield); mp 112−114 °C; *R_f_* = 0.6 (petroleum ether); ^1^H NMR (600 MHz, CDCl_3_) *δ* 7.48−7.41 (m, 4H) and 7.05−6.98 (m, 4H) ppm; HRMS (ESI) m/z [M]^+^ calculated for C_12_H_8_F_2_S_2_ 254.0035, found 254.0029; ^19^F NMR (376 MHz, CDCl_3_) *δ* -113.4 ppm.

**Bis(3-fluorophenyl) disulfide (3k) [75]**. White solid (20.6 mg, 81% yield); mp 93−94 °C; *R_f_* = 0.6 (Petroleum ether); ^1^H NMR (600 MHz, CDCl_3_) *δ* 7.31−7.27 (m, 2H), 7.26 (s, 1H), 7.25−7.23 (m, 2 H), 7.22 (*t*, *J* = 1.8 Hz, 1H), and 6.96−6.91 (m, 2H) ppm; HRMS (ESI) m/z [M]^+^ calculated for C_12_H_8_F_2_S_2_ 254.0035, found 254.0030; ^19^F NMR (376 MHz, CDCl_3_) *δ* -111.1 ppm.

**Bis(2-fluorophenyl)disulfide (3l)** [75]. Slight yellow oil (19.9 mg, 75% yield); *R_f_* = 0.6 (petroleum ether); ^1^H NMR (600 MHz, CDCl_3_) *δ* 7.59 (*td*, *J* = 7.8, 1.8 Hz, 2 H), 7.28−7.25 (m, 2H), 7.12 (*td*, *J* = 7.8, 1.2 Hz, 2 H), and 7.08−7.04 (m, 2H) ppm; HRMS (ESI) m/z [M]^+^ calculated for C_12_H_8_F_2_S_2_ 254.0035, found 254.0029; ^19^F NMR (376 MHz, CDCl_3_) *δ* -109.9 ppm.

**4,4’-Dichlorodiphenyl disulfide (3p)** [73]. White solid (26.0 mg, 91% yield); mp 68−70 °C; *R_f_* = 0.6 (petroleum ether); ^1^H NMR (600 MHz, CDCl_3_) *δ* 7.40 (*dt*, *J* = 8.4, 4.8 Hz, 4H) and 7.28 (*dt*, *J* = 8.4, 4.8 Hz, 4H) ppm; ^13^C{^1^H} NMR (100 MHz, CDCl_3_) *δ* 135.1, 133.7, 129.4, and 129.3; HRMS (ESI) m/z [M]^+^ calculated for C_12_H_8_Cl_2_S_2_ 285.9444, found 285.9421.

**Bis(3-clorophenyl)disulfide (3q)** [73]. White solid (25.2 mg, 88% yield); mp 80−82 °C; *R_f_* = 0.6 (petroleum ether); ^1^H NMR (600 MHz, CDCl_3_) *δ* 7.48 (*d*, *J* = 2.4 Hz, 1 H), 7.37−7.32 (m, 2 H), 7.26 (*d*, *J* = 2.4 Hz, 1H), and 7.24−7.20 (m, 4H) ppm; HRMS (ESI) m/z [M]^+^ calculated for C_12_H_8_Cl_2_S_2_ 285.9444, found 285.9425.

**Bis(2-clorophenyl)disulfide (3r)** [76]. White solid (24.0 mg, 84% yield); mp 90−91 °C; *R_f_* = 0.6 (petroleum ether); ^1^H NMR (400 MHz, CDCl_3_) *δ* 7.48 (*dd*, *J* = 8.0, 1.6 Hz, 2H), 7.30 (*dd*, *J* = 7.2, 1.6 Hz, 2H), 7.14 (td, *J* = 7.6, 1.2 Hz, 2H), and 7.09 (*td*, *J* = 7.6, 1.6 Hz, 2H) ppm; HRMS (ESI) m/z [M]^+^ calculated for C_12_H_8_Cl_2_S_2_ 285.9444, found 285.9421.

**Bis(4-bromophenyl)disulfide (3s)** [75]. White solid (33.3 mg, 89% yield); mp 110−112 °C; *R_f_* = 0.6 (petroleum ether); ^1^H NMR (600 MHz, CDCl_3_) *δ* 7.44−7.41 (m, 4 H) and 7.35−7.32 (m, 4 H) ppm; HRMS (ESI) m/z [M]^+^ calculated for C_12_H_8_Br_2_S_2_ 373.8434, found 373.8414.

**2,2’-Dinaphthyl disulfide (3t)** [77]. White solid (26.1 mg, 82% yield); mp 139−141 °C; *R_f_* = 0.6 (petroleum ether); ^1^H NMR (600 MHz, CDCl_3_) *δ* 7.99 (*d*, *J* = 1.8 Hz, 2H), 7.82−7.76 (m, 4H), 7.75−7.71 (m, 2H), 7.62 (*dd*, *J* = 9.0, 6.6 Hz, 2H), and 7.49−7.44 (m, 4H) ppm; HRMS (ESI) m/z [M + H]^+^ calculated for C_20_H_15_S_2_ 319.0610, found 319.0614.

**Bis(4-trifluoromethylphenyl)disulfide (3u)** [42]. White solid (28.3 mg, 80% yield); mp 119−120 °C; R*_f_* = 0.6 (petroleum ether); ^1^H NMR (600 MHz, CDCl_3_) *δ* 7.60−7.55 (m, 8H) ppm; HRMS (ESI) m/z [M + H]^+^ calculated for C_14_H_9_F_6_S_2_ 355.0044, found 355.0039; ^19^ F NMR (376 MHz, CDCl_3_) *δ* -62.5 ppm.

## 4. Conclusions

In summary, we have developed an efficient protocol for the selective access to symmetrical diaryl sulfides and disulfides using sodium sulfinates as sulfenylation reagents via homocoupling reaction. The utilization of readily available sodium sulfinates as coupling partners and good functional group tolerance with modest to excellent yields for most substrates enable these two types of novel transformations to become attractive alternatives for the preparation of the corresponding sulfur compounds. More importantly, sodium sulfinates were used for the first time to access symmetrical diaryl sulfides. The convinced mechanism, selectivity, and synthetic application of this transformation are still under investigation.

## Data Availability

Not applicable.

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
