# Peer review of "Homocouplings of Sodium Arenesulfinates: Selective Access to Symmetric Diaryl Sulfides and Diaryl Disulfides"

_molecules, 2022, doi:10.3390/molecules27196232_

Round 1
Reviewer 1 Report
I find the work new and can be improved with the following additions.
1) can author present a characteristic analysis of sulfide and disulfide compounds, if not please add a IR spectrum of at least one compound from sulfide and one with disfide bonds.
2) I would suggest validating the strategy with intramolecular cyclization of the sulfide reaction, that would certainly improve the novelty and application of the process. 3) The only advancement in the work would be a transformation of sulfinate to diaryl sulfides. It would be a more useful strategy if the authors can provide clear mechanistic evidence on the exact pathway, so that in future this could be applied by other researchers across the globe. 4) may be for future studies, using a second metal catalyst to activate the other counter coupling reagent authors may achieve the unsymmetrical-sulfides.Author Response
1.1) can author present a characteristic analysis of sulfide and disulfide compounds, if not please add a IR spectrum of at least one compound from sulfide and one with disulfide bonds.
Thanks for the reviewer’s question. In order to discriminate sulfide and disulfide products, HPLC analysis was carried out for diphenyl sulfide and disulfide, and the retention times were different obviously. Furthermore, the IR spectrum of diphenyl sulfide and diphenyl disulfide were also determined, while the spectrum had less difference with each other.
Both the HPLC data and IR spectrum have been added in the revised manuscript.
1.2) I would suggest validating the strategy with intramolecular cyclization of the sulfide reaction, that would certainly improve the novelty and application of the process.
Thanks for the reviewer’s suggestion. The bissulfinates was thus prepared and subjected to the standard conditions of the sulfide reaction, and the desired product dibenzothiophene was obtained in 51% yield.
1.3) The only advancement in the work would be a transformation of sulfinate to diaryl sulfides. It would be a more useful strategy if the authors can provide clear mechanistic evidence on the exact pathway, so that in future this could be applied by other researchers across the globe.
Thanks for the reviewer’s suggestion. We truly agree with the reviewer’s opinion about the mechanism. However, there’s no further evidences to prove the mechanism at this stage, and we will continue doing some control experiments for the mechanistic study in the future.
1.4) may be for future studies, using a second metal catalyst to activate the other counter coupling reagent authors may achieve the unsymmetrical-sulfides.
Thanks for the reviewer’s suggestion. Actually, a second metal catalyst such as copper and silver was also tried for the synthesis of unsymmetric sulfides, while the desired product was not detected.

Reviewer 2 Report
The article presents the material in a consistent and accessible way. The Introduction contains enough informative sources of literature. The reaction conditions were well chosen by the authors and series of symmetrical sulfides and disulfides were obtained.
There is a proposal to add a review: Synthesis of Thioethers from Sulfonyl Chlorides, Sodium Sulfinates, and Sulfonyl Hydrazides Synthesis 2019; 51(19): 3567-3587.
Note: Not all compounds contain 13C spectra?
What ratio of solvents is selected for chromatography?
Author Response
Response to Reviewer 2:
2.1) There is a proposal to add a review: Synthesis of Thioethers from Sulfonyl Chlorides, Sodium Sulfinates, and Sulfonyl Hydrazides Synthesis 2019; 51(19): 3567-3587.
Thanks for the reviewer’s suggestion. The mentioned review paper has been added as Ref.3h in the revised manuscript.
2.2) Not all compounds contain 13C spectra?
Thanks for the reviewer’s question. Because some of products are known compounds, just the 1H NMR spectrum were done and they were consistent with the reported data. Therefore, only the unknown compounds were given with both 1H and 13C NMR in the supporting information.
2.3) What ratio of solvents is selected for chromatography?
Thanks for the reviewer’s reminder. The ratio of solvents has been added in the revised manuscript.

Round 2
Reviewer 1 Report
can the authors add the spectra and experimental data in both manuscript and supporting info for the cyclic product 2t.
please label the IR spectra with the characterstic peak for the disulfide bond.
Author Response
1.1 can the authors add the spectra and experimental data in both manuscript and supporting info for the cyclic product 2t.
Thanks for the reviewer’s reminder. The 1HNMR spectra and the characterization of 2t has been added in the revised manuscript and supporting information.
1.2 please label the IR spectra with the characterstic peak for the disulfide bond.
Thanks for the reviewer’s suggestion. We have searched some literature and IR data to discriminate the disulfide bond, while the S-S stretching vibration at 500-400 cm-1 was very weak according to the Sadtler Handbook of Infrared Spectra. Notably, the C-S stretching (600-700) and Ph-S deformation (1400-1000) vibration, which both exist in diphenyl sulfide and diphenyl disulfide were labeled.